# Analysis of clinicopathological factors associate with the visibility of early gastric cancer in endoscopic examination and usefulness of linked color imaging: A multicenter prospective study

**Kensuke Fukuda**[1], **Kazuhiro Mizukami**[1]*, **Daisuke Yamaguch**[2], **Yuichiro Tanaka**[2], **Kazutoshi Hashiguchi**[3], **Takashi Akutagawa**[4], **Ryo Shimoda**[4], **Sho Suzuki**[5], **Tadashi Miike**[5], **Yorinobu Sumida**[6], **Hidehito Maeda**[7], **Fumisato Sasaki**[7], **Ryosuke Gushima**[8], **Hideaki Miyamoto**[8], **Keiichi Hashiguchi**[9], **Naoyuki Yamaguchi**[9], **Tetsuya Ohira**[10], **Tetsu Kinjo**[10], **Ken Ohnita**[11], **Tomohiko Moriyama**[12], **Kensei Ohtsu**[13], **Akira Aso**[14], **Ryo Ogawa**[1], **Tetsuya Ueo**[15], **Masahide Fukuda**[1]

1 Department of Gastroenterology, Faculty of Medicine, Oita University, Oita, Japan, 2 Department of Gastroenterology, National Hospital Organization Ureshino Medical Center, Saga, Japan, 3 Interventional Endoscopy Center, Josuikai Imamura Hospital, Saga, Japan, 4 Department of Endoscopic Diagnostics and Therapeutics, Saga University Hospital, Saga, Japan, 5 Division of Endoscopy and Center for Digestive Disease, Department of Gastroenterology and Hepatology, University of Miyazaki Hospital, Miyazaki, Japan, 6 Department of Gastroenterology, Kitakyushu Municipal Medical Center, Fukuoka, Japan, 7 Department of Digestive and Lifestyle Diseases, Kagoshima University Graduate School of Medical and Dental Sciences, Kagoshima, Japan, 8 Department of Gastroenterology and Hepatology, Kumamoto University Hospital, Kumamoto, Japan, 9 Department of Endoscopy and Gastroenterology, Nagasaki University Hospital, Nagasaki, Japan, 10 Department of Endoscopy, Ryukyu University Hospital, Okinawa, Japan, 11 Department of Gastroenterology and Hepatology, Shunkaikai Inoue Hospital, Nagasaki, Japan, 12 International Medical Department, Kyushu University, Fukuoka, Japan, 13 Department of Gastroenterology, Tobata Kyoritsu Hospital, Fukuoka, Japan, 14 Aso Clinic, Fukuoka, Japan, 15 Department of Gastroenterology, Oita Red Cross Hospital, Oita, Japan

* mizkaz0809@oita-u.ac.jp

## Abstract

### Background

This study investigated clinicopathological factors associated with the visibility of early gastric cancer and the efficacy of linked color imaging.

### Methods

Patients with early gastric cancer who underwent endoscopic treatment between April 2021 and July 2022 were enrolled. All cases underwent white light imaging and linked color imaging. Three experts evaluated lesion visibility using a visual analog scale. A mean score ≥3 on white light imaging was defined as "good visibility", and <3 as "poor visibility". We extracted patient information and endoscopic and pathological data for the lesion and background mucosa, analyzed factors associated with the visibility of early gastric cancer, and compared visibility between white light imaging and linked color imaging.

**Data Availability Statement:** All relevant data are within the paper and its Supporting Information files.

**Funding:** The author(s) received no specific funding for this work.

**Competing interests:** The authors have declared that no competing interests exist.

## Results

Ninety-seven lesions were analyzed, with good visibility in 49 and poor visibility in 48. Multivariate analysis revealed small lesion size (odds ratio 1.89) and presence of endoscopic intestinal metaplasia (odds ratio 0.49) as significantly associated with the poor visibility of early gastric cancer. Mean visibility score was significantly higher for linked color imaging (P<0.001). Mean score for linked color imaging was significantly higher in the poor visibility group (P<0.001), but not significantly different in the good visibility group (P = 0.292). Mean score was significantly higher with linked color imaging in cases with endoscopic intestinal metaplasia (P = 0.0496) and lesions <20 mm in diameter (<10 mm, P = 0.002; 10–20 mm, P = 0.004).

## Conclusions

Lesion size and endoscopic intestinal metaplasia are associated with the visibility of early gastric cancer in white light imaging. Linked color imaging improves visibility of gastric cancer with these factors.

## Introduction

Although the number of gastric cancer cases and associated deaths are decreasing with the widespread use of *Helicobacter pylori* (*H. pylori*) eradication therapy, gastric cancer remains the sixth most common cancer worldwide and the fourth leading cause of cancer-related death [1]. Early endoscopic detection of gastric cancer is essential for reducing its mortality rate of gastric cancer [2, 3]. Gastric cancers occurring among *H. pylori*-uninfected patients and after *H. pylori* eradication have increased, and it has been noted that these gastric cancers have an endoscopic appearance that is different from the conventional differentiated adenocarcinoma commonly observed in *H. pylori*-infected patients [4–6].

Gastric cancers after *H. pylori* eradication are also more difficult to visualize endoscopically than conventional gastric adenocarcinomas. In the gastric mucosa after *H. pylori* eradication, diffuse redness disappears due to the recovery from inflammation and atrophy, and a map-like appearance of mottled, patchy redness appears [7]. Further, gastric cancer after *H. pylori* eradication is characterized by endoscopic findings such as redness, flat depressed lesions, and small size [8, 9] and histological features include the presence of non-neoplastic mucosa covering the tumor surface, as a "gastritis-like appearance" [10]. For these reasons, detecting gastric cancer after *H. pylori* eradication may be more difficult [11].

What clinicopathological characteristics of gastric cancer make endoscopic detection difficult? Although some studies have examined the characteristics of each gastric cancer type, no study have comprehensively examined the association between endoscopic visibility of gastric cancer and patients' background, endoscopic findings, pathological findings. The primary endpoint of this study was to identify clinicopathological factors associated with the visibility of early gastric cancer on endoscopy.

In addition, Linked color imaging (LCI) is an image-enhanced endoscopy that has been shown to improve the visibility of gastrointestinal neoplasms compared to white light imaging (WLI) [12, 13]. We therefore investigated whether LCI can improve visibility for these lesions.

## Methods

### Study subjects

This multicenter, prospective study was conducted from April 30, 2021 to July 31, 2022 by GI-Kyushu, a clinical research organization involving 15 high-volume facilities in the Kyushu region of Japan. All subjects provided written informed consent for inclusion before participating in the study. The study was conducted in accordance with the Declaration of Helsinki, and the protocol was approved by the ethics committees of Oita University (No. 1928) and registered with the University Hospital Medical Information Network (UMIN No. 000042069). Patients aged >20 years old with early flat-type gastric cancer (0-IIa, IIb, or IIc) ≤30 mm who consented to participate were consecutively enrolled in the study. Patients with gastric cancer that obviously invaded the submucosa and was not indicated for endoscopic treatment or postoperative stomach, or were deemed inappropriate by the responsible physician were excluded from this study.

### Endoscopic equipment

The LASEREO/ELUXEO system (light source LL-7000/BL-7000, processor VP-7000; Fujifilm Corp., Tokyo, Japan) and a high-definition monitor were used. The LCI mode of this system acquires high-contrast blood vessel information and rich color information by irradiating short-wavelength narrow-band light and white light, enabling enhancement of slight differences in mucosal color by extending saturation and hue differences of colors close to mucosal color.

### Endoscopic imaging

Endoscopy was performed by 18 endoscopy experts, each with over 12 years of experience. Endoscopic observation was performed in accordance with the recommendations in the quality assurance manual of endoscopic screening for gastric cancer in Japanese communities [14]. Endoscopic images were taken at the following 20 locations with WLI and LCI: antegrade view of upper, middle, lower, posterior wall of gastric body, posterior wall of upper gastric angle, four-quadrant views of gastric antrum, retroflexed view of lesser curvature of gastric angle, lesser curvature of lower, middle, upper gastric body, antegrade view of anterior wall and greater curvature of lower, middle, upper gastric body, retroflexed view of cardia, posterior wall of lower gastric body, antegrade view of posterior wall of gastric angle, prepylorus. Twenty images of WLI and LCI per case were taken with the same composition (S1 Fig). These images did not focus on the target lesion. At least one of the 20 images should include the lesion. All endoscopic images were collected from participating centers at a central adjudication facility, randomized by WLI and LCI, and then sent to three experts for visibility assessment.

### Assessment of lesion visibility

Three endoscopic experts (22, 23, and 28 years of endoscopic experience, respectively) from the participating institutions were selected to evaluate the visibility of lesions. They were not involved in any endoscopic examinations related to this study. Evaluators were blinded to all patient and lesion information, except that the number of lesions was only informed to evaluators if a case had multiple lesions. Three endoscopists independently searched for lesions in 20 randomly arranged screening images and evaluated their visibility using the five-point visual analog scale (VAS) ranging from 1(poor: not detectable or detected wrong site) to 5(detectable at a glance). Visibility was assessed by both WLI and LCI, with the order of searching randomly determined for each case. The results of the evaluations by each endoscopists were sent

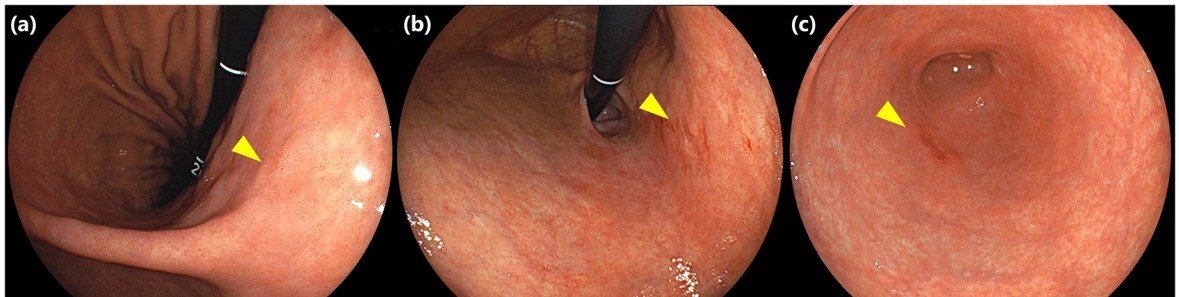

**Fig 1. Representative images of early gastric cancer with mean visibility score of 1.0, 3.0, 5.0 on WLI.** (a) Type 0-IIb early gastric cancer located on posterior wall of gastric angle (yellow triangle). Mean visibility score of this lesion was 1.0. (b) Type 0-IIc early gastric cancer located on lesser curvature of the upper gastric body (yellow triangle). Mean visibility score of this lesion was 3.0. (c) Type 0-IIc early gastric cancer located on greater curvature of gastric antrum (yellow triangle). Mean visibility score of this lesion was 5.0.

to a central adjudication facility and collected. Representative images of lesions with different visibility scores on WLI are shown in Fig 1. Lesions with mean visibility score ≥3 by 3 endoscopists on WLI showed "good visibility" and lesions with mean visibility score <3 on WLI showed "poor visibility".

## Patient and endoscopic information

The following background information was collected for patients: age, sex, underlying disease, medication, and *H. pylori* infection status. *H. pylori* infection status was defined as follows. Patients without endoscopic atrophy (C0/1) and with negative results for the rapid urease test (RUT) and urea breath test (UBT) were considered uninfected. Current infection was considered present if at least one of the RUT or UBT was positive regardless of the degree of endoscopic atrophy. Patients with history of eradication or endoscopic atrophy (≥C2) but negative RUT and UBT were considered to show past infection. The following information about the lesion and background mucosa was evaluated and registered by endoscopists at each institution as endoscopic findings. For lesions, location, size, color, and morphology were registered. For background mucosa, endoscopic atrophy, intestinal metaplasia (IM), and map-like redness were evaluated using the Kyoto classification of gastritis [15]. Endoscopic atrophy was evaluated based on the Kimura–Takemoto classification [16]: C-0, none; C-1 or C-2, mild; C-3 or O-1, moderate; and O-2 or O-3, severe. The following endoscopic findings on WLI were used as indicators of IM: whitish mucosa, rough or uneven mucosal surface, and villous appearance. A previous study found that when these endoscopic findings on WLI were defined as IM, the diagnostic accuracy of IM improved with sensitivity of 94.6%, specificity of 69.1%, and ROC/AUC of 0.818 [17]. The degree of IM was classified as none, patchy, partial, or total.

## Histological evaluation

All early gastric cancers enrolled in this study were treated by ESD. Tumor size, histology, invasion depth, and proportion of epithelium with low-grade atypia (ELA) were evaluated in the resected specimen. According to previous reports [18, 19], ELA was pathologically defined as low-grade atypia or normal columnar epithelium covering the cancer surface. The degree of ELA was calculated as the number of sections showing ELA divided by the number of sections of the lesion: 0%, none; ≤33%, mild; 34–66%, moderate; and ≥67%, severe. Pathological evaluation of gastric background mucosa was performed using the normal mucosal area of the specimen removed by ESD. Histological assessment of inflammation, neutrophil activity, atrophy, and IM of surrounding mucosa were based on the updated Sydney System [20]. In three of the

12 facilities (Nagasaki University, Kumamoto University, and Inoue Hospital), only 12 lesions were evaluated by expert pathologists at each center. The remaining resection specimens were collected at Oita University and evaluated by an expert pathologist.

### Statistical analysis

Continuous variables such as patient age and lesion size were expressed as median and range. Univariate and multivariate analyses using logistic regression analyses were performed to examine clinicopathological factors related to lesion visibility, factors that were significant in the univariate analysis were imputed to the multivariate analysis. Interobserver agreement for WLI and LCI was calculated using Fleiss's kappa. The Wilcoxon signed-rank sum test was used to compare visibility scores between WLI and LCI. All statistical analyses were performed using EZR software (Saitama Medical Center, Jichi Medical University, Saitama, Japan). Values of P<0.05 were considered significant.

## Results

### Baseline characteristics of patients, early gastric cancer, and background gastric mucosa

Between April 2021 and July 2022, 110 lesions (108 patients) were enrolled. Of these, 4 lesions >3 cm, 6 adenomas, 1 type 0-I lesion, and 2 lesions with insufficient patient/lesion information were excluded. Ninety-seven lesions in 96 patients were then analyzed, with 49 lesions in the good visibility group and 48 lesions in the poor visibility group (Fig 2). Interobserver agreement for lesion visibility by three evaluators was 0.526 for WLI and 0.504 for LCI. Baseline characteristics of patients, lesions, and background mucosa are shown in Table 1. *H. pylori* infection status was uninfected in 8.2%, current in 21.6%, and past infection in 70.1%. Median lesion size was 11.0 mm, 70 lesions (72.2%) were reddish in color, 66 lesions (68%) were depressed type, and 79 lesions (81.4%) were well-differentiated adenocarcinoma. The background mucosa showed moderate or severe atrophy in 83 lesions (85.6%) and endoscopic IM in 72 lesions (74.2%).

### Univariate and multivariate analysis of factors associated with visibility of early gastric cancer on WLI

Of the 97 lesions, 49 were classified as good visibility group and 48 as poor visibility group. Only small lesion size (odds ratio [OR] 1.89, 95% confidence interval [CI] 1.07–3.34) and presence of endoscopic IM (OR 0.52, 95%CI 0.29–0.93) were significantly associated with poor visibility of lesion on WLI in univariate analyses (P = 0.03, P = 0.03, respectively). Visibility was not associated with any pathological factors, including degree of ELA. Multivariate analysis identified small lesion size (OR 1.93, 95%CI 1.07–3.46; P = 0.03) and presence of endoscopic IM (OR 0.49, 95%CI 0.27–0.91; P = 0.02) as factors independently associated with poor visibility of lesion on WLI (Table 2).

### Comparison of lesion visibility on WLI and LCI

Lesion visibility was better with LCI (median, 3.3 [Interquartile Range (IQR), 2.3–4.0]) than with WLI (median, 3.0 [IQR, 1.7–4.0]) (P<0.001) (Fig 3). Fig 4 shows representative images of a lesion that was classified as poor visibility on WLI but was improved to good visibility on LCI. In the good visibility group, visibility did not differ significantly between WLI (median, 4.0 [IQR, 3.3–4.5]) and LCI (median, 4.0 [IQR, 3.7–4.5]) (P = 0.29). In the poor visibility

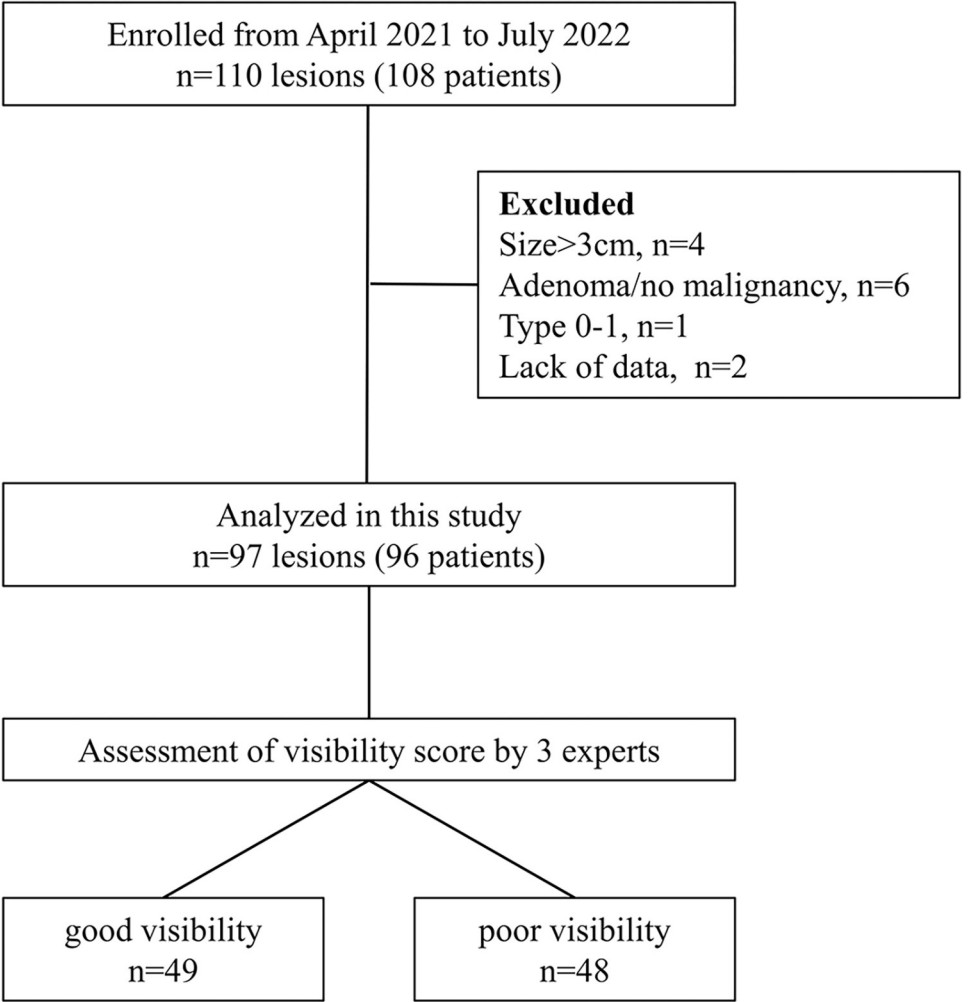

**Fig 2. Study flowchart.** Good visibility, lesions with mean visibility score ≥3 on WLI; poor visibility, lesions with mean visibility score <3 on WLI.

group, visibility was significantly better with LCI (median, 2.3 [IQR, 1.3–3.0]) than with WLI (median, 1.6 [IQR, 1.3–2.3]) (P<0.001) (Fig 5).

In lesions without IM (n = 25), no significant difference in mean visibility score was seen between WLI (median, 3.7 [IQR, 2.7–4.3]) and LCI (median, 3.7 [IQR, 3.0–4.3]) (P = 0.29). In lesions with IM (n = 72), visibility was clearly better with LCI (median, 3.2 [IQR, 1.9–4.0]) than with WLI (median, 2.7 [IQR, 1.7–3.7])) (P = 0.0496). Regarding lesion size, another factor associated with visibility, visibility scores did not differ significantly between groups for lesions 21–30 mm (WLI median, 3.7 [IQR, 3.0–4.3]) vs. LCI median, 3.7 [IQR, 3.3–4.5]); P = 0.40), but were significantly better with LCI (median, 3.0 [IQR, 1.7–3.7])) than with WLI (median, 2.3 [IQR, 1.7–3.1]) (P = 0.002) for lesions <10 mm and with LCI (median, 3.3 [IQR, 2.4–4.3]) than with WLI (median, 3.0 [IQR, 1.7–4.3]) (P = 0.004) for lesions 11–20 mm (Fig 6).

## Discussion

This is the first study that comprehensively examined factors affecting endoscopic visibility for early gastric cancer, including not only endoscopic and pathological findings of gastric cancer

**Table 1. Baseline characteristics of patients and early gastric cancer and background gastric mucosa.**

|  | n | (%) |
|---|---|---|
| **Total number of lesions** | 97 |  |
| **Age (year), median (IQR)** | 73.0 (43–92) |  |
| **Sex** |  |  |
| Male | 77 | 79.4 |
| Female | 20 | 20.6 |
| **Comorbidities** |  |  |
| Hypertension | 45 | 46.4 |
| Diabetes | 20 | 20.6 |
| Reflux esophagitis | 7 | 7.2 |
| Other organ cancer | 14 | 14.4 |
| **Medicine** |  |  |
| Anti-acid drugs | 30 | 30.9 |
| Low-dose aspirin | 16 | 16.5 |
| Anticoagulants | 10 | 10.3 |
| Prednisolone | 3 | 3.1 |
| NSAIDs | 3 | 3.1 |
| **_H. pylori_** |  |  |
| Uninfected | 8 | 8.2 |
| Current infection | 21 | 21.6 |
| Past infection | 68 | 70.1 |
| **Tumor location** |  |  |
| U | 19 | 19.6 |
| M | 38 | 39.2 |
| L | 40 | 41.2 |
| **Color** |  |  |
| Whitish | 16 | 16.5 |
| Normal tone | 11 | 11.3 |
| Redness | 70 | 72.2 |
| **Macroscopic type** |  |  |
| 0-IIa | 19 | 19.6 |
| 0-IIb | 9 | 9.3 |
| 0-IIc | 66 | 68.0 |
| Others | 3 | 3.1 |
| **Tumor size (mm), median (IQR)** | 11.0 (2–30) |  |
| **Histological type** |  |  |
| Undifferentiated | 6 | 6.2 |
| Moderately differentiated | 12 | 12.4 |
| Well-differentiated | 79 | 81.4 |
| **Endoscopic atrophy** |  |  |
| None | 5 | 5.2 |
| Mild (C-1, C-2) | 9 | 9.3 |
| Moderate (C-3, O-1) | 31 | 32.0 |
| Severe (O-2, O-3) | 52 | 53.6 |
| **Endoscopic intestinal metaplasia** |  |  |
| None | 25 | 25.8 |
| Patchy | 18 | 18.6 |
| Partial | 29 | 29.9 |

(_Continued_)

**Table 1.** (Continued)

|  | n | (%) |
|---|---|---|
| Overall | 25 | 25.8 |
| **Map-like redness** |  |  |
| None | 58 | 59.8 |
| Patchy | 7 | 7.2 |
| Partial | 26 | 26.8 |
| Overall | 6 | 6.2 |

Data were presented as number (percentage) of patients unless otherwise indicated. Data for age and tumor size are presented as median. IQR, interquartile range; NSAIDs, non-steroidal anti-inflammatory drugs.

and background mucosa but also, patient background and *H. pylori* infection status. We also investigated the effectiveness of LCI on the visibility of early gastric cancer. The results showed that lesion size and endoscopic IM are associated with the visibility of early gastric cancer in WLI and LCI improves visibility of gastric cancer with negative visibility factors.

The characteristics of gastric cancer have changed, with the recent increases in gastric cancer after *H. pylori* eradication, the recognition of *H. pylori*-uninfected gastric cancer, and the establishment of new concepts. Gastric endoscopic screening is currently becoming more and more difficult to perform.

In the present study, the incidence rate of gastric cancer after eradication was overwhelmingly high. Previous studies have shown that *H. pylori* eradication suppresses the incidence of gastric cancer, but a certain percentage of gastric cancers still occur after eradication [21, 22]. Since the incidence rate of gastric cancer after *H. pylori* eradication is expected to increase with the spread of *H. pylori* eradication therapy, gastric endoscopic screening that contributes to gastric cancer detection after eradication will be required.

In the present study, endoscopic IM was an independent factor in making gastric cancer difficult to visualize. There are no studies that investigated the relationship between endoscopic findings of the gastric mucosa and the visibility of gastric cancer. IM is found in the course of chronic gastritis and atrophic gastritis caused by *H. pylori* infection and has been considered a risk factor for gastric cancer [23–25]. IM is characterized by morphological features such as villous appearance, whitish mucosa, and rough mucosal surface on WLI [17]. In addition, after *H. pylori* eradication, IM is characterized by the appearance of patchy, map-like redness, representing depressed erythema of various sizes [7]. In this study, type 0-I lesions were excluded as clearly identifiable, mostly type 0-II lesions with slight elevations or depressions, and frequently an erythematous tone. In general, early gastric cancer after *H. pylori* eradication is often differentiated adenocarcinoma, which endoscopically presents as small, erythematous, depressed lesions [26]. Early gastric cancer with subtle morphology and coloration may be intermingled in the gastric background mucosa with IM, reducing lesion visibility.

In this study, multivariate analysis showed the visibility of early gastric cancer worsened with decreasing size. Few papers have examined lesion size and visibility on WLI. A previous study revealed that small gastric tumors less than one cm in size were a risk factor of missed lesion during 12 months follow up after endoscopic resection of gastric tumors [27]. This result supports the results of the present study but included not only gastric cancer but also adenoma. The other study reported no significant difference in visibility comparing early gastric cancers ≥21 mm and ≤20 mm (OR 1.1, 95%CI 0.96–1.26; P = 0.161) [28]. Unlike previous studies, this study evaluated lesion visibility by identifying lesions in 20 images. This method

**Table 2.** Univariate and multivariate analysis of factors associated with visibility of early gastric cancer on WLI.

| | Univariate analysis | | | Multivariate analysis | | |
|---|---|---|---|---|---|---|
| | OR | 95%Cl | P-Value | OR | 95%Cl | P-Value |
| **Age** | 1.01 | 0.96–1.06 | 0.73 | | | |
| **Male sex** | 0.617 | 0.23–1.68 | 0.34 | | | |
| **Comorbidities** | | | | | | |
| Hypertension | 1.05 | 0.47–2.32 | 0.91 | | | |
| Diabetes | 0.449 | 0.16–1.25 | 0.13 | | | |
| Reflux esophagitis | 1.33 | 0.28–6.30 | 0.72 | | | |
| Other organ cancer | 1.89 | 0.58–6.13 | 0.29 | | | |
| **Medicine** | | | | | | |
| anti acid drugs | 0.8 | 0.34–1.90 | 0.61 | | | |
| low dose aspirin | 0.98 | 0.33–2.85 | 0.96 | | | |
| anticoagulant | 0.38 | 0.09–1.57 | 0.18 | | | |
| Prednisolone | 0.48 | 0.04–5.47 | 0.55 | | | |
| NSAIDs | 0 | 0.04-Inf | 0.99 | | | |
| ***H. pylori*** | | | | | | |
| uninfection | 0.98 | 0.23–4.16 | 0.98 | | | |
| current infection | 1.41 | 0.53–3.72 | 0.49 | | | |
| past infection | 0.77 | 0.32–1.83 | 0.55 | | | |
| **Tumor location** | | | | | | |
| U | 0.86 | 0.31–2.33 | 0.76 | | | |
| M | 0.81 | 0.36–1.84 | 0.62 | | | |
| L | 1.36 | 0.60–3.06 | 0.46 | | | |
| **Color** | | | | | | |
| Normal tone | 0.52 | 0.14–1.91 | 0.33 | | | |
| Redness | 1.4 | 0.57–3.42 | 0.46 | | | |
| Whitish | 0.98 | 0.33–2.85 | 0.96 | | | |
| **Macroscopic type** | | | | | | |
| 0-IIa | 1.23 | 0.48–3.20 | 0.67 | | | |
| 0-IIb | 1.25 | 0.32–4.97 | 0.76 | | | |
| 0-IIc | 0.84 | 0.35–2.03 | 0.70 | | | |
| **Tumor size (0<10mm, 1<20mm, 2≦30mm)** | 1.89 | 1.07–3.34 | 0.03* | 1.93 | 1.07–3.46 | 0.03* |
| **Endoscopic atrophy** | 0.96 | 0.60–1.54 | 0.87 | | | |
| none | 0.64 | 0.10–4.00 | 0.63 | | | |
| mild (C-1, C-2) | 1.25 | 0.32–4.97 | 0.75 | | | |
| moderate (C-3, O-1) | 1.29 | 0.55–3.04 | 0.56 | | | |
| severe (O-2, O-3) | 0.81 | 0.36–1.80 | 0.61 | | | |
| **Endoscopic intestinal metaplasia** | 0.52 | 0.29–0.93 | 0.03* | 0.49 | 0.27–0.91 | 0.02* |
| none | 3.4 | 1.26–9.15 | 0.02* | | | |
| patchy/partial | 0.64 | 0.29–1.41 | 0.27 | | | |
| overall | 0.56 | 0.22–1.42 | 0.23 | | | |
| **Maplike redness** | 0.78 | 0.46–1.30 | 0.34 | | | |
| none | 1.9 | 0.83–4.32 | 0.13 | | | |
| patchy/partial | 0.51 | 0.22–1.19 | 0.12 | | | |
| overall | 0.98 | 0.19–5.11 | 0.08 | | | |
| **Histological type** | 0.78 | 0.38–1.61 | 0.50 | | | |
| Well differentiated | 0.59 | 0.21–1.68 | 0.32 | | | |
| Moderately differentiated | 2.15 | 0.60–7.67 | 0.24 | | | |

*(Continued)*

**Table 2.** (Continued)

| | Univariate analysis | | | Multivariate analysis | | |
|---|---|---|---|---|---|---|
| | OR | 95%Cl | P-Value | OR | 95%Cl | P-Value |
| Undifferentiated | 0.98 | 0.19–5.11 | 0.98 | | | |
| **Depth of tumor (M/SM1 vs SM2)** | 3.07 | 0.31–30.60 | 0.34 | | | |
| **ELA (non/mild vs moderate/severe)** | 0.74 | 0.33–1.69 | 0.48 | | | |
| **Histological atrophy** | 1.34 | 0.59–3.06 | 0.48 | | | |
| **Histological IM** | 1.13 | 0.51–2.51 | 0.77 | | | |

Uni- and multivariate analyses using logistic regression analysis were performed to examine clinicopathological factors related to lesion visibility. WLI, White light imaging; OR, odds ratio; CI, confidence interval; NSAIDs, non-steroidal anti-inflammatory drugs; M, tumor confined to the mucosa or invasion into the muscularis mucosa; SM1, tumor invasion less than 0.5 mm into the submucosa; SM2, tumor invasion of 0.5 mm or more into the submucosa; ELA, epithelium with low-grade atypia * Statistically significant.

assumes a situation consistent with actual clinical gastric screening endoscopy, and the results suggested that lesion size affects visibility during routine endoscopy.

Conversely, the degree of ELA was not associated with lesion visibility in this study. ELA represents a coating of tumor cells by epithelial cells with low or no cellular atypia [11, 18, 19]. It has been thought that gastric cancer after eradication, which is characterized by ELA, is more difficult to detect endoscopically than *H. pylori* current infection or uninfected gastric cancer. However, there are no study that clarify this. This study suggested that the degree of ELA and *H. pylori* infection status are not related to the visibility of gastric cancer.

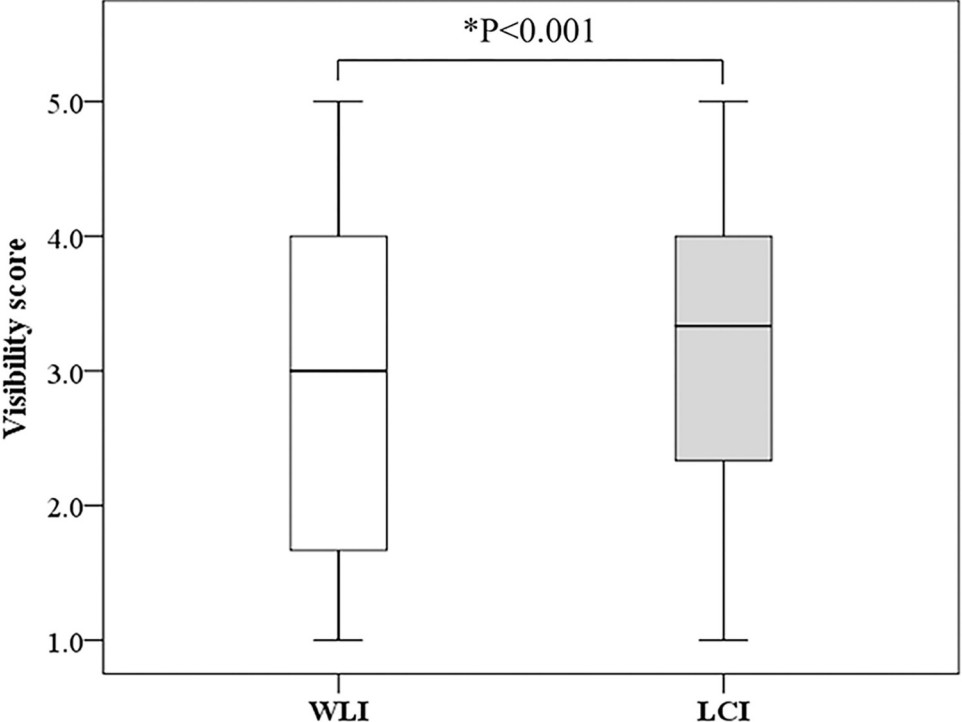

**Fig 3.** Comparison of mean visibility score between WLI and LCI. WLI, White light imaging; LCI, Linked color imaging; *, statistically significant.

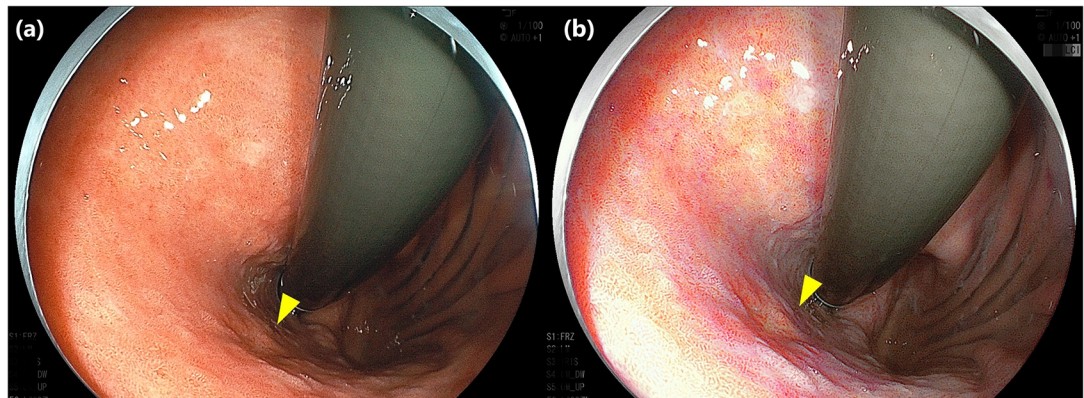

**Fig 4. Representative images of early gastric cancer in which the visibility was classified as poor visibility on WLI but improved to good visibility on LCI.** Type 0-IIc early gastric cancer, 9 mm in size, located on posterior wall of the upper gastric body (yellow triangle). The mean visibility score on WLI was 1.7, which was classified as poor visibility (a), but the mean visibility score on LCI improved to 3.0 (b). WLI, White light imaging; LCI, Linked color imaging.

The current study showed that endoscopic gastric screening with LCI improves gastric cancer visibility. LCI is an image-enhanced endoscopy that uses two types of narrow-band laser light to rearrange the color information obtained with narrow-band light and white light to enhance the tonal difference between reddish and whitish tones. In the diagnosis of early gastric cancer, LCI has also been shown to improve lesion visibility by significantly increasing the color difference between the lesion and surrounding mucosa [13, 29], and LCI has been shown to improve lesion visibility compared to WLI in endoscopic gastric screening [12, 30, 31]. The present results, as in previous studies, showed significantly improved lesion visibility scores for LCI compared to WLI. In particularly, the present study found no significant difference between LCI and WLI in lesions with good visibility on WLI. LCI significantly improved visibility in lesions with poor visibility on WLI. This suggests that LCI may facilitate detection for lesions that are difficult for endoscopists to find. Observations of LCI also showed improved visibility scores in lesions with IM. Previous studies have reported that IM exhibits purple called "lavender color" when observed using LCI [32]. It has also been shown that LCI

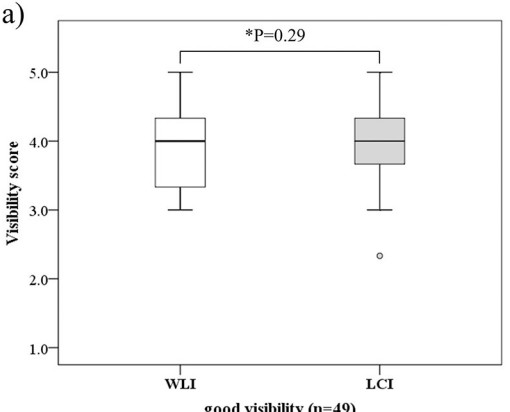

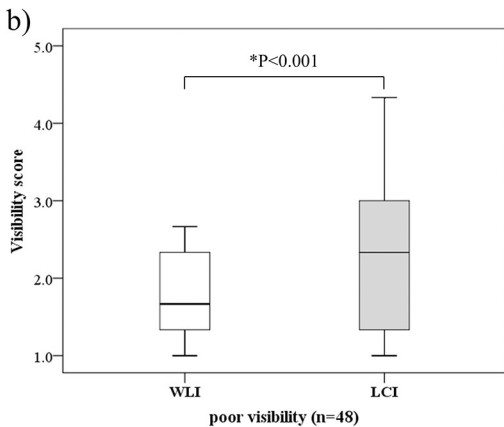

**Fig 5. Comparison of mean visibility score of WLI and LCI in good visibility lesions and poor visibility lesions.** WLI, White light imaging; LCI, Linked color imaging; good visibility, lesions with mean visibility score ≥3 on WLI; poor visibility, lesions with mean visibility score <3 on WLI; *, statistically significant.

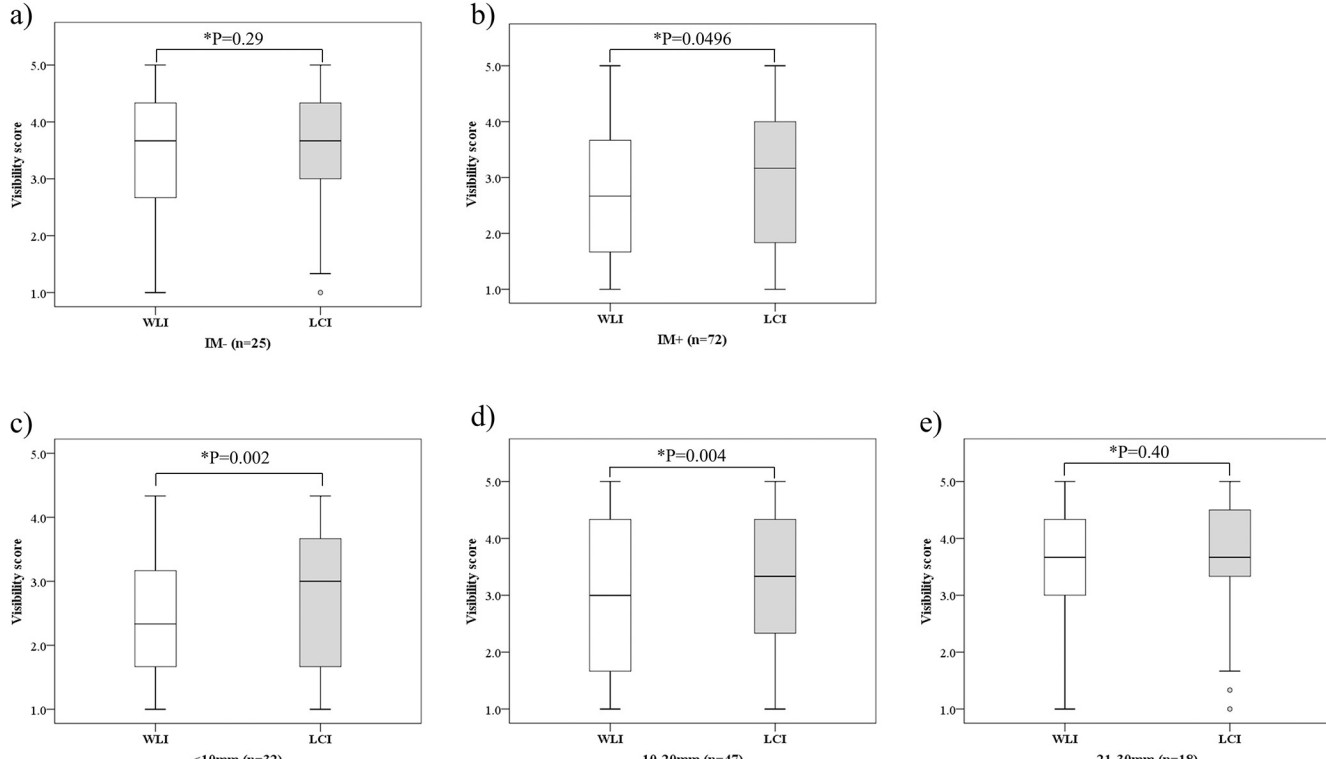

**Fig 6. Comparison of visibility score of WLI and LCI by presence or absence of intestinal metaplasia and lesion size.** (a), (b) Comparison of visibility score of WLI and LCI by presence or absence of intestinal metaplasia, (c), (d), (e) Comparison of visibility score of WLI and LCI by lesion size. WLI, White light imaging; LCI, Linked color imaging; IM-, lesions without endoscopic intestinal metaplasia; IM+, lesions with mild or sever endoscopic intestinal metaplasia; *, statistically significant.

increases the color difference between the lesion and the surrounding mucosa even in lesions surrounded by intestinal metaplasia [29]. The color difference between the lesion and purple-colored IM is enhanced, presumably increasing lesion visibility. As a factor that reduced gastric cancer visibility in this study, IM is expected to prove advantageous in LCI due to its characteristics.

Furthermore, no difference in visibility was seen between WLI and LCI for relatively large lesions >20 mm, but visibility was significantly improved for LCI compared to WLI for small lesions <10 mm and 11–20 mm. From these results, LCI can be inferred to provide significant benefits not only for normal lesions, but also even for difficult-to-see lesions (lesions with IM, lesions <20 mm, etc.) that might have been overlooked by WLI. In this study, there were 58 cases of lesions <20mm with IM, and in 35 (60%) of these, visibility was improved with LCI compared to WLI. These results suggest that early gastric cancer with severe IM in the background gastric mucosa and small size, which are difficult to detect on WLI, can be easier to detect using LCI.

This study showed several limitations. First, pathological evaluation of background gastric mucosa was not performed according to the Updated Sydney system. In this study, endoscopic IM was related to lesion visibility, but histological IM was uninvolved. This discrepancy in IM between endoscopy and histology remains problematic. The Updated Sydney system recommends biopsy of the antrum and corpus to accurately assess histological atrophy and IM [20]. However, in this study, we did not perform additional biopsy to avoid invasiveness to the patient, and instead used a non-tumor area of resected specimens to evaluate the histology of

the background gastric mucosa. This suggests that the severity of histological IM may not be accurately evaluated and may be the cause of the discrepancy between endoscopic IM and histological IM. Second, this study used still images to evaluate lesion visibility. Since lesions were identified from 20 screening images that were not focused on the lesion, the distance from the endoscope varies for each lesion. Lesions far from the endoscope may appear dark, and lesions close to the scope may be affected by halation, which may affect visibility. Furthermore, the visibility evaluation using still images did not consider the time required to detect lesions. In actual clinical practice, the time required for endoscopic examination is limited, and lesions that require time to be detected have low visibility and are more likely to be overlooked. Therefore, the time required to detect a lesion is an important factor in evaluating the visibility of a lesion. If visibility is to be evaluated in a situation that assume actual clinical setting, visibility should have been evaluated in real time while performing an endoscopic examination, or it should have been evaluated using endoscopic video rather than endoscopic images. Third, the sample size was small for the number of items considered. Finally, visibility determinations were performed by endoscopic specialists, but may be performed by unskilled endoscopists in routine practice, thus contributing to bias. Because of the present study was a pilot study, further large-scale clinical studies involving young endoscopists should be conducted assuming various endoscopic gastrointestinal screening conditions, and this will hopefully lead to the development of screening strategies for new gastric lesions. In conclusion, with WLI, smaller lesion size and more severe endoscopic IM were associated with lower visibility of early gastric cancer, and with LCI, inclusion of these lesions improved the visibility of early gastric cancer.

## Supporting information

**S1 Fig. Twenty endoscopic images recommended in quality assurance manual of endoscopic screening for gastric cancer in Japanese community.** Endoscopic images was obtained for each WLI and LCI in upper, middle, lower, posterior wall of gastric body (a-c), posterior wall of upper gastric angle (d), four-quadrant views of gastric antrum (e-h), lesser curvature of gastric angle (i), lesser curvature of lower, middle, upper gastric body (j-l), anterior wall and greater curvature of lower, middle, upper gastric body(m-p), retroflexed view of cardia, posterior wall of lower gastric body (q,r), antegrade view of posterior wall of gastric angle, prepylorus (s,t). These images were not focused on the target lesion and at least one of the 20 images should include the lesion.
(TIF)

**S1 File.**
(XLSX)

## Author Contributions

**Conceptualization:** Kazuhiro Mizukami.

**Data curation:** Kensuke Fukuda, Kazuhiro Mizukami.

**Formal analysis:** Kensuke Fukuda, Kazuhiro Mizukami.

**Investigation:** Kensuke Fukuda, Kazuhiro Mizukami, Daisuke Yamaguch, Yuichiro Tanaka, Kazutoshi Hashiguchi, Takashi Akutagawa, Ryo Shimoda, Sho Suzuki, Tadashi Miike, Yorinobu Sumida, Hidehito Maeda, Fumisato Sasaki, Ryosuke Gushima, Hideaki Miyamoto, Keiichi Hashiguchi, Naoyuki Yamaguchi, Tetsuya Ohira, Tetsu Kinjo, Ken Ohnita, Tomohiko Moriyama, Akira Aso, Ryo Ogawa, Tetsuya Ueo, Masahide Fukuda.

**Methodology:** Kensuke Fukuda, Kazuhiro Mizukami, Daisuke Yamaguch, Kazutoshi Hashiguchi, Takashi Akutagawa, Ryo Shimoda, Sho Suzuki, Tadashi Miike, Yorinobu Sumida, Hidehito Maeda, Fumisato Sasaki, Ryosuke Gushima, Hideaki Miyamoto, Keiichi Hashiguchi, Naoyuki Yamaguchi, Tetsuya Ohira, Tetsu Kinjo, Ken Ohnita, Tomohiko Moriyama, Kensei Ohtsu, Akira Aso.

**Project administration:** Kazuhiro Mizukami.

**Supervision:** Kazuhiro Mizukami.

**Validation:** Kensuke Fukuda, Kazuhiro Mizukami, Daisuke Yamaguch, Ryo Shimoda, Tadashi Miike, Hidehito Maeda, Fumisato Sasaki, Keiichi Hashiguchi, Kensei Ohtsu.

**Writing – original draft:** Kensuke Fukuda.

**Writing – review & editing:** Kensuke Fukuda, Kazuhiro Mizukami, Daisuke Yamaguch, Yuichiro Tanaka, Kazutoshi Hashiguchi, Takashi Akutagawa, Ryo Shimoda, Sho Suzuki, Tadashi Miike, Yorinobu Sumida, Hidehito Maeda, Fumisato Sasaki, Ryosuke Gushima, Hideaki Miyamoto, Keiichi Hashiguchi, Naoyuki Yamaguchi, Tetsuya Ohira, Tetsu Kinjo, Ken Ohnita, Tomohiko Moriyama, Kensei Ohtsu, Akira Aso, Ryo Ogawa, Tetsuya Ueo, Masahide Fukuda.

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
