## [Decision Letter · Decision Letter 0]

9 Aug 2024

PONE-D-24-24128Analysis of clinicopathological factors associate with the visibility of early gastric cancer in endoscopic examination and usefulness of Linked color imagingPLOS ONE

Dear Dr. Mizukami,

Thank you for submitting your manuscript to PLOS ONE. After careful consideration, we feel that it has merit but does not fully meet PLOS ONE’s publication criteria as it currently stands. Therefore, we invite you to submit a revised version of the manuscript that addresses the points raised during the review process.

When responding to reviewers comments, please make sure to detail the following aspects of your work; More clarity on the visibility scores of early gastric cancer using endoscopic imagesMore clarity regarding assessment of lesion visibility by Endoscopists and their selection criteriaDetails on the gastric screening endoscopic examination.==============================

Please submit your revised manuscript by Sep 23 2024 11:59PM. If you will need more time than this to complete your revisions, please reply to this message or contact the journal office at plosone@plos.org. Please include the following items when submitting your revised manuscript:A rebuttal letter that responds to each point raised by the academic editor and reviewer(s). You should upload this letter as a separate file labeled 'Response to Reviewers'.A marked-up copy of your manuscript that highlights changes made to the original version. You should upload this as a separate file labeled 'Revised Manuscript with Track Changes'.An unmarked version of your revised paper without tracked changes. You should upload this as a separate file labeled 'Manuscript'.

We look forward to receiving your revised manuscript.

Kind regards,

Elingarami Sauli, PhD

Academic Editor

PLOS ONE

Journal Requirements:

Reviewers' comments:

Reviewer's Responses to Questions

**Comments to the Author**

1. Is the manuscript technically sound, and do the data support the conclusions?

Reviewer #1: Partly

Reviewer #2: Yes

Reviewer #3: Yes

Reviewer #4: Yes

Reviewer #5: Yes

2. Has the statistical analysis been performed appropriately and rigorously? 

Reviewer #1: No

Reviewer #2: Yes

Reviewer #3: Yes

Reviewer #4: No

Reviewer #5: Yes

3. Have the authors made all data underlying the findings in their manuscript fully available?

Reviewer #1: Yes

Reviewer #2: Yes

Reviewer #3: Yes

Reviewer #4: Yes

Reviewer #5: Yes

4. Is the manuscript presented in an intelligible fashion and written in standard English?

Reviewer #1: Yes

Reviewer #2: Yes

Reviewer #3: Yes

Reviewer #4: Yes

Reviewer #5: Yes

5. Review Comments to the Author

Reviewer #1: In the method section, “Patients with gastric cancer > 30mm, morphological type 0-I gastric cancer”, and “who did not consent to participate” are antonyms of the inclusion criteria already stated and therefor unnecessary as exclusion criteria and should be removed.

How many sites participated in this study should be described in the method section.

Were the endoscopic images collected from the participating sites to the central adjudication site?

The visibility score that authors described in this paper is not a Visual Analogue Scale which is a quantifiable numerical rating using a linear scale, but a kind of ordinal qualitative scale.

How were three experts who evaluated the visibility score chosen? Whether were they blinded from the enrolled patients and corresponding images of WLI and LCI ?

Who evaluated the endoscopic findings other than visibility score, endoscopists at each site?

How were the values for imputing to the logistic regression model chosen?

The comparison of visibility score between WLI and LCI was analyzed using Wilcoxon signed rank test, so to avoid misinterpretation, median with IQR should be stated and presented as a box plot with p-value. As an alternative test method, it may be appropriate to use McNemar test to compare the proportion of at least a score of 3 between WLI and LCI.

Comparing visibility scores between WLI and LCI in subgroups of good visibility or poor visibility is inappropriate because good or poor visibility were outcomes, not background characteristics of lesions. Figure 4 and lines 264 to 267 should be removed.

For patients with two lesions, how was each lesion assessed? Was it possible to analyze the lesions separately to assess the visibility score? Was the number of lesions informed to the assessor?

Figures 1 and 2 are similar, therefore Figure 1 should be removed.

Table 2 is an analysis of the lesions on WLI. The authors should make that clear in the table title and manuscript.

Lines 215 to 216 appeared to be factually incorrect.

Lines 220 to 221 are hard to understand why there is a contradiction between the process of eligible patients enrollment and the high rate of eradication of H. pylori.

The improved detection of early gastric cancer with LCI described in lines 279 to 281 should be suggested by the cases with improved visibility scores. How many cases have had improved visibility scores with LCI when the lesions were small and involved IM?

Reviewer #2: Overall:

The authors examined the association between endoscopic visibility of gastric cancer and patients’ background, endoscopic findings, pathological findings. Moreover, they investigated whether LCI can improve visibility for these lesions. They showed that lesion size and endoscopic IM are associated with the visibility of early gastric cancer in WLI and LCI improves visibility of gastric cancer with negative visibility factors.

Comments:

1. Please describe and explain interobserver agreement of expert endoscopists.

2. If the authors have cases in which WLI had better visibility than LCI, please describe and explain.

Reviewer #3: Thank you for the opportunity to review this article. There are a few points we need to clarify before accepting the article, so here is a list of those points.

Major

1．In the Introduction and Discussion sections, the authors emphasize that the visibility of gastric cancer after H. pylori （HP） eradication is poor and the detection rate is low. However, why did the current study include cases of both current infection and uninfected HP? If the authors want to resolve a clinical question, it may be better to limit the study to cases after HP eradication.

2. Please clarify the number of years of experience in endoscopic practice of the three endoscopists who evaluated the endoscopic images.

3. VAS-based evaluation is a subjective evaluation and lacks objectivity. Is there a correlation between "VAS-based evaluation" and "the actual color difference obtained from WLI and LCI images"? I think it is important to confirm that there is a correlation between the subjective evaluation and the color difference.

4. How long did it take to detect the lesion after viewing the endoscopic images of the 20 pairs? If you are assuming a situation consistent with actual clinical gastric screening endoscopic examinations, please clarify this because it is an important point.

Minor

1. The resolution of the image is poor, so the text cannot be seen in detail.

Reviewer #4: The authors conducted a multicenter prospective study to investigate the clinicopathological findings that are useful in identifying gastric cancers. Moreover, the authors investigate the usefulness of Linked Color imaging. The manuscript is written comprehensively, however, there are some concerns that should be addressed.

Major

1. Title. Please clearly indicate in the title that the present study was conducted prospectively in multicenter institutions.

2. Abstract. Although the authors describe the odds ratio to show the results, I think it is better to describe the direction of the results clearly. For example, lesion size is associated with “good” visibility, while intestinal metaplasia is associated with “bad” visibility.

3. Introduction. The authors described that “Early endoscopic detection of gastric cancer is essential for reducing its mortality rate of gastric cancer”, please add the reference for this sentence.

4. Introduction and Methods. The authors emphasize the difficulty of identifying gastric cancers after H.pylori eradication in the introduction section. Moreover, the authors described the characteristic findings of gastric cancers after H. Pylori eradication. Moreover, the authors also described in the 1st paragraph of the introduction section that the findings of gastric cancers after H. pylori eradication are different from those of H. pylori-infected gastric cancers. However, the authors included three types of gastric cancers in the analysis. I think it is better to focus on gastric cancers after H.pylori eradication. The situation where endoscopists do not know or cannot judge H. pylori's status is highly unlikely. Information about H.pylori infection status is critically important when conducting upper screening endoscopy.

5. Methods. Regarding the assessment of lesion visibility, did three endoscopists evaluate the lesion separately? Moreover, how did they evaluate the images? Did they use a cloud-type database? How much time did they have to evaluate the lesion? Was the final decision made by a majority vote of those three endoscopists?

6. Statistical analysis. Were there any measurements for sample size calculation?

7. Results. Regarding the paragraph on page 12 which showed the results of univariate analysis, please describe the direction of the results as I pointed out in the abstract.

8. Results. Please describe the number of good or poor visibility patients in the manuscript. (Although, the authors showed it in the figure). This information is important for logistic regression analysis.

9. Discussion. The “rate” in the following sentence could be revised as“incidence rate”. “The present study enrolled as many early gastric cancers £30 mm that met the eligibility criteria as possible, but the rate of gastric cancer after eradication was overwhelmingly high.” “Since the rate of gastric cancer after H. pylori eradication is expected to increase with the spread of H. pylori eradication therapy, gastric endoscopic screening that contributes to gastric cancer detection after eradication will be required.”

Reviewer #5: This manuscript describes the clinicopathological factors associated with the visibility of early gastric cancer and the efficacy of linked color imaging.

I read this manuscript with great interest. I have a few comments as shown below.

1. Please provide a representative images evaluated "poor visibility" by white light but "good visibility" by LCI.

2. p 18, line 256

In the discussion, the authors state "The current study showed that endoscopic gastric screening with image-enhanced endoscopy improves gastric cancer visibility". This sentence can be interpreted to mean that all IEEs can improve the visibility of cancer. The results of screening endoscopy with other IEEs are not discussed in the manuscript. Is there any evidence that other image-enhanced endoscopy also improves gastric cancer visibility? If so, please describe it, and if there is no clear evidence, just describe the LCI results used in this study.

3. This study evaluates visibility by looking at recorded endoscopic images. Please include in the appropriate part of the discussion that a clinical study to evaluate the real-time detection rate of gastric cancer during endoscopic examination is desired.

6. PLOS authors have the option to publish the peer review history of their article (what does this mean?). If published, this will include your full peer review and any attached files.

Reviewer #1: No

Reviewer #2: No

Reviewer #3: No

Reviewer #4: No

Reviewer #5: No

---

## [Author Response · Author response to Decision Letter 0]

25 Sep 2024

Dear Editor and reviewers

Thank you very much for reviewing our manuscript and offering valuable advice. We have carefully reviewed the comments and revised the manuscript accordingly. Our responses are given in a point-by-point manner below. Changes to the manuscript are shown in underline/ red bold.

We hope the revised version is now suitable for publication and look forward to hearing from you in due course.

Sincerely,

Kazuhiro Mizukami

Response to Reviewer #1

We thank the reviewer for positive comments and their valuable feedback which have helped us significantly improve the paper. We have answered each of your points below.

1) In the method section, “Patients with gastric cancer > 30mm, morphological type 0-I gastric cancer”, and “who did not consent to participate” are antonyms of the inclusion criteria already stated and therefor unnecessary as exclusion criteria and should be removed.

R: We thank the reviewer for these comments. We completely agree that three exclusion criteria reviewer mentioned are antonyms of the inclusion criteria and are unnecessary. We have removed the relevant sections in lines 61 to 64.

Lines 61-64; Patients with gastric cancer >30 mm, morphological type 0-I gastric cancer, gastric cancer that obviously invaded the submucosa and was not indicated for endoscopic treatment or postoperative stomach, or who did not consent to participate or were deemed inappropriate by the responsible physician were excluded from this study.

2) How many sites participated in this study should be described in the method section.

R: We appreciate reviewer’s comment. We have added the number of participating facilities to lines 55 as follows.

Lines 55; a clinical research organization involving 15 high-volume facilities

3) Were the endoscopic images collected from the participating sites to the central adjudication site?

R: We appreciate reviewer’s comment. As reviewer pointed out, endoscopic images collected from the participating sites to the central adjudication site. The endoscopic images were then sent to three endoscopic experts. We have added the sentence to lines 84-86 as below.

Lines 84-86 All endoscopic images were collected from participating centers at a central adjudication facility, randomized by WLI and LCI, and then sent to three experts for visibility assessment.

4) The visibility score that authors described in this paper is not a Visual Analogue Scale which is a quantifiable numerical rating using a linear scale, but a kind of ordinal qualitative scale.

R: We appreciate reviewer’s comment. As the reviewer pointed out, the Visual Analog Scale described in the paper looks like an ordinal qualitative scale. This is probably because the scale was defined to make it easier for the readers to understand. In the actual evaluation, the evaluators were asked to score the item at their own discretion between 1 (poor: not detectable, detected wrong site) and 5 (detectable at a glance). In response to the reviewer's suggestion, we have changed lines 96 to 98 as follows:

Lines 95-97: the five-point visual analog scale (VAS) ranging from 1(poor: not detectable or detected wrong site) to 5(detectable at a glance).

5) How were three experts who evaluated the visibility score chosen? Whether were they blinded from the enrolled patients and corresponding images of WLI and LCI?

R: We appreciate reviewer’s comment. Three experienced endoscopists from the participating institutions were randomly selected to evaluate the visibility. They were not involved in any endoscopic examinations related to this study. Evaluators were blinded to all patient and lesion information, except that the number of lesions was only informed to evaluators if a case had multiple lesions. We have added following sentence to lines 90 to 94.

Lines 90-94: Three endoscopic experts (22, 23, and 28 years of endoscopic experience, respectively) from the participating institutions were selected to evaluate the visibility. They were not involved in any endoscopic examinations related to this study. Evaluators were blinded to all patient and lesion information, except that the number of lesions was only informed to evaluators if a case had multiple lesions.

6) Who evaluated the endoscopic findings other than visibility score, endoscopists at each site?

R: We appreciate reviewer’s comment. That is true. Endoscopic findings other than lesion visibility were evaluated by endoscopists at each site. We have changed Lines 119 to120 as below.

Lines 118-119: The following information about the lesion and background mucosa was evaluated and registered by endoscopists at each institution as endoscopic findings.

7) How were the values for imputing to the logistic regression model chosen?

R: We appreciate reviewer’s comment. Factors that were significant in the univariate analysis were imputed to the multivariate analysis. We have added the same sentence to lines 148 to 149.

Lines 147-148: factors that were significant in the univariate analysis were imputed to the multivariate analysis.

8) The comparison of visibility score between WLI and LCI was analyzed using Wilcoxon signed rank test, so to avoid misinterpretation, median with IQR should be stated and presented as a box plot with p-value. As an alternative test method, it may be appropriate to use McNemar test to compare the proportion of at least a score of 3 between WLI and LCI.

R: We appreciate reviewer’s comment. As pointed out by the reviewer, we made mistakes in the description of the data and figures. Since we used the Wilcoxon singed rank test to compare of visibility score between WLI and LCI, we should have stated the median and IQR. In response to the comment, we changed the (mean score) to (median [IQR]) in lines 198-213 in the manuscript and changed the figure 3,5 and 6. Thank you very much for your accurate comment.

9) Comparing visibility scores between WLI and LCI in subgroups of good visibility or poor visibility is inappropriate because good or poor visibility were outcomes, not background characteristics of lesions. Figure 4 and lines 264 to 267 should be removed.

R: We appreciate reviewer’s comment. We agree with the reviewer's comment. However, We believe that the results of this study, that visibility was equivalent between WLI and LCI in lesions with good visibility on WLI, and that visibility was improved with LCI in lesions with poor visibility on WLI, are meaningful. This result also supports the statement in the discussion (lines 293) that "This suggests that LCI may facilitate detection for lesions that are difficult for endoscopists to find." If the reviewer could allow us to include this analysis result in the text, we would be very grateful.

10) For patients with two lesions, how was each lesion assessed? Was it possible to analyze the lesions separately to assess the visibility score? Was the number of lesions informed to the assessor?

R: We appreciate reviewer’s comment. As we answered in the fifth question, when there were multiple lesions, we informed only the number of lesions to avoid confusion for the evaluator. The evaluators assessed visibility of each lesion separately. In fact, there was only one case in this study with two lesions.

11) Figures 1 and 2 are similar, therefore Figure 1 should be removed.

R: We appreciate reviewer’s comment. We completely agree. We have changed Lines 55-56 as follow and removed Figure 1 and Figure legend of manuscript.

Lines 55-56; a clinical research organization involving 15 high-volume facilities in the Kyushu region of Japan, using the method shown in Fig 1.

12) Table 2 is an analysis of the lesions on WLI. The authors should make that clear in the table title and manuscript.

R: We appreciate reviewer’s comment. We completely agree. We have emphasized in lines 179 to 180 and 187to 188 and table title (lines 189). 

Lines 179-180: Univariate and multivariate analysis of factors associated with visibility of early gastric cancer on WLI

Lines 187-188: independently associated with poor visibility of lesion on WLI (Table 2).

Lines 189-190: Table 2: Univariate and multivariate analysis of factors associated with visibility of early gastric cancer on WLI.

13) Lines 215 to 216 appeared to be factually incorrect.

R: We appreciate reviewer’s comment. We completely agree. I think it's an insufficient expression and incorrect. We have removed that section.

Lines 244-245; Previously, gastrointestinal endoscopic screening only had to look for differentiated gastric cancer associated with H. pylori.

14) Lines 220 to 221 are hard to understand why there is a contradiction between the process of eligible patients enrollment and the high rate of eradication of H. pylori.

R: We appreciate reviewer’s comment. We completely agree. There is no reason why the expression in the text is a contradiction, and it seems difficult to understand. Therefore, part of the text has been chaged as follows

Lines 248-249; In the present study, enrolled as many early gastric cancers �30 mm that met the eligibility criteria as possible, but the incidence rate of gastric cancer after eradication was overwhelmingly high.

15) The improved detection of early gastric cancer with LCI described in lines 279 to 281 should be suggested by the cases with improved visibility scores. How many cases have had improved visibility scores with LCI when the lesions were small and involved IM?

R: We appreciate reviewer’s comment. As the reviewer says, the effectiveness of LCI would be clearer if the number of cases in which visibility was improved with LCI was listed. In this study, there were 58 cases of lesions less than 2 cm in size with IM, and in 35 (60%) of these, visibility was improved with LCI compared to WLI. We have added the same sentence to lines 305 to 307.

Lines 305-307; In this study, there were 58 cases of lesions <20mm with IM, and in 35 (60%) of these, visibility was improved with LCI compared to WLI.

Response to Reviewer #2

Overall

The authors examined the association between endoscopic visibility of gastric cancer and patients’ background, endoscopic findings, pathological findings. Moreover, they investigated whether LCI can improve visibility for these lesions. They showed that lesion size and endoscopic IM are associated with the visibility of early gastric cancer in WLI and LCI improves visibility of gastric cancer with negative visibility factors.

R: We wish to express our appreciation to the Reviewer for insightful comments, which have helped us significantly improve the paper. We have answered each of your points below.

1) Please describe and explain interobserver agreement of expert endoscopists.

R: We appreciate reviewer’s comment. Following the reviewer's suggestion, we calculated the interobserver agreement using the Fleiss’ Kappa. The results were as follow:

κ=0.526 (WLI), κ=0.504 (LCI)

Based on the above results, the following sentence was added to the manuscript:

Lines 148-149; Interobserver agreement for WLI and LCI was calculated using Fleiss's kappa.

Lines 161-162; Interobserver agreement for lesion visibility by three evaluators was 0.526 for WLI and 0.504 for LCI.

2) If the authors have cases in which WLI had better visibility than LCI, please describe and explain.

R: We appreciate reviewer’s comment. Of the 97 lesions, 16 lesion (16%) showed better visibility with WLI compared to LCI. The clinicopathological characteristics of these 16 lesions were not identified. However, among the 16 lesions, some were imaged at close range, and halation appeared to affect the visibility of the lesions. We thought that the effect of halation was greater in LCI, which may have led to a negative evaluation of visibility in LCI. In this study, the 20 screening images were not focused on the lesions, so the distance from the endoscope varies for each lesion. Lesions far from the endoscope may appear darker, and lesions close to the endoscope may be affected by halation. In response to the reviewer's comments, we thought that evaluating visibility with still images was one of the limitations of this study, so we added the following sentence to the lines 319 to 329 of the Discussion. Thank you very much for very important comments.

Lines 319-329; Second, this study used still images to evaluate lesion visibility. Since lesions were identified from 20 screening images that were not focused on the lesion, the distance from the endoscope varies for each lesion. Lesions far from the endoscope may appear dark, and lesions close to the scope may be affected by halation, which may affect visibility. Furthermore, the visibility evaluation using still images did not consider the time required to detect lesions. In actual clinical practice, the time required for endoscopic examination is limited, and lesions that require time to be detected have low visibility and are more likely to be overlooked. Therefore, the time required to detect a lesion is an important factor in evaluating the visibility of a lesion. If visibility is to be evaluated in a situation that assume actual clinical setting, visibility should have been evaluated in real time while performing an endoscopic examination, or it should have been evaluated using endoscopic video rather than endoscopic images.

Response to Reviewer #3

Thank you for the opportunity to review this article. There are a few points we need to clarify before accepting the article, so here is a list of those points.

R: We wish to express our appreciation to the Reviewer for insightful comments, which have helped us significantly improve the paper. We have answered each of your points below.

Major

1) In the Introduction and Discussion sections, the authors emphasize that the visibility of gastric cancer after H. pylori （HP） eradication is poor and the detection rate is low. However, why did the current study include cases of both current infection and uninfected HP? If the authors want to resolve a clinical question, it may be better to limit the study to cases after HP eradication.

R: We appreciate reviewer’s comment. As the reviewer pointed out, gastric cancer that occurs after H. pylori eradication has been considered to be more difficult to detect by endoscopy than gastric cancer with current H. pylori infection because of changes in the background gastric mucosa, such as map-like redness, and the surface of the lesion being covered by normal mucosa, but there have been no papers that objectively prove this. Therefore, as described in lines 41 to 46, our aim in this study was to comprehensively examine all clinical pathological factors, including the infection status of H. pylori, that affect the visibility of gastric cancer. Therefore, this study included not only patients after H. pylori eradication, but also uninfected and current infection. 

2) Please clarify the number of years of experience in endoscopic practice of the three endoscopists who evaluated the endoscopic images.

R: We appreciate reviewer’s comment. The three endoscopists had 22, 23, and 28 years of endoscopic experience, respectively. We have added following sentence to lines 91 to 92.

Lines 90-91; Three endoscopic experts (22, 23, and 28 years of endoscopic experience, respectively) from the participating institutions were selected to evaluate the visibility of lesions.

3) VAS-based evaluation is a subjective evaluation and lacks objectivity. Is there a correlation between "VAS-based evaluation" and "the actual color difference obtained from WLI and LCI images"? I think it is important to confirm that there is a correlation between the subjective evaluation and the color difference.

R: We appreciate reviewer’s comment. As the reviewer pointed out, the color difference between the lesion and the surrounding mucosa is one of the important factors related to the visibility of the lesion. Therefore, as the reviewer suggested, the validity of the subjective visibility assessment may be proven by examining the relationship between the VAS-based evaluation and the actual color difference, but the actual visibility evaluation may be affected not only by the color difference but also by morphological changes such as the unevenness of the surrounding mucosa due to atrophy and intestinal metaplasia

---

## [Decision Letter · Decision Letter 1]

7 Oct 2024

Analysis of clinicopathological factors associate with the visibility of early gastric cancer in endoscopic examination and usefulness of Linked color imaging: a multicenter prospective study

PONE-D-24-24128R1

Dear Dr. Mizukami,

We’re pleased to inform you that your manuscript has been judged scientifically suitable for publication and will be formally accepted for publication once it meets all outstanding technical requirements.

Kind regards,

Elingarami Sauli, PhD

Academic Editor

PLOS ONE

Additional Editor Comments (optional):

All the comments from original reviewers, including myself, have been very well addressed.

Reviewers' comments:

Reviewer's Responses to Questions

**Comments to the Author**

1. If the authors have adequately addressed your comments raised in a previous round of review and you feel that this manuscript is now acceptable for publication, you may indicate that here to bypass the “Comments to the Author” section, enter your conflict of interest statement in the “Confidential to Editor” section, and submit your "Accept" recommendation.

Reviewer #1: All comments have been addressed

Reviewer #2: (No Response)

Reviewer #3: All comments have been addressed

Reviewer #6: All comments have been addressed

2. Is the manuscript technically sound, and do the data support the conclusions?

Reviewer #1: Yes

Reviewer #2: Yes

Reviewer #3: Yes

Reviewer #6: Yes

3. Has the statistical analysis been performed appropriately and rigorously? 

Reviewer #1: Yes

Reviewer #2: Yes

Reviewer #3: Yes

Reviewer #6: Yes

4. Have the authors made all data underlying the findings in their manuscript fully available?

Reviewer #1: Yes

Reviewer #2: Yes

Reviewer #3: Yes

Reviewer #6: Yes

5. Is the manuscript presented in an intelligible fashion and written in standard English?

Reviewer #1: Yes

Reviewer #2: Yes

Reviewer #3: Yes

Reviewer #6: Yes

6. Review Comments to the Author

Reviewer #1: (No Response)

Reviewer #2: The authors examined the association between endoscopic visibility of gastric cancer and patients’ background, endoscopic findings, pathological findings.

The authors have appropriately revised the paper.

Reviewer #3: The authors have responded sincerely to the reviewers' comments, and the manuscript has been revised well.

I think this manuscript will be acceptable.

Reviewer #6: The manuscript has been revised in response to the reviewer's comments (some parts that cannot be revised are noted in the "Limitations" section), and there are no additional comments.

7. PLOS authors have the option to publish the peer review history of their article (what does this mean?). If published, this will include your full peer review and any attached files.

Reviewer #1: No

Reviewer #2: **Yes: **Kazuo Yashima

Reviewer #3: No

Reviewer #6: **Yes: **Yoriaki Komeda

---

## [Editor Report · Acceptance letter]

25 Oct 2024

PONE-D-24-24128R1 

PLOS ONE

Dear Dr. Mizukami, 

I'm pleased to inform you that your manuscript has been deemed suitable for publication in PLOS ONE. Congratulations! Your manuscript is now being handed over to our production team.

Kind regards, 

on behalf of

Dr. Elingarami Sauli 

Academic Editor

PLOS ONE